# Structural Phase Transition and Metallization of Nanocrystalline Rutile Investigated by High-Pressure Raman Spectroscopy and Electrical Conductivity

**Meiling Hong [1,2], Lidong Dai [1,*] , Heping Li [1], Haiying Hu [1], Kaixiang Liu [1,2], Linfei Yang [1,2] and Chang Pu [1,2]**

[1] Key Laboratory of High-Temperature and High-Pressure Study of the Earth's Interior, Institute of Geochemistry, Chinese Academy of Sciences, Guiyang 550081, China
[2] University of Chinese Academy of Sciences, Beijing 100049, China
*   Correspondence: dailidong@vip.gyig.ac.cn; Tel.: +86-851-8555-9715

**Abstract:** We investigate the structural, vibrational, and electrical transport properties of nanocrystalline rutile and its high-pressure polymorphs by Raman spectroscopy, and *AC* complex impedance spectroscopy in conjunction with the high-resolution transmission electron microscopy (HRTEM) up to ~25.0 GPa using the diamond anvil cell (DAC). Experimental results indicate that the structural phase transition and metallization for nanocrystalline rutile occurred with increasing pressure up to ~12.3 and ~14.5 GPa, respectively. The structural phase transition of sample at ~12.3 GPa is confirmed as a baddeleyite phase, which is verified by six new Raman characteristic peaks. The metallization of the baddeleyite phase is manifested by the temperature-dependent electrical conductivity measurements at ~14.5 GPa. However, upon decompression, the structural phase transition from the metallic baddeleyite to columbite phases at ~7.2 GPa is characterized by the inflexion point of the pressure coefficient and the pressure-dependent electrical conductivity. The recovered columbite phase is always retained to the atmospheric condition, which belongs to an irreversible phase transformation.

**Keywords:** nanocrystalline rutile; phase transition; metallization; high pressure; diamond anvil cell

## 1. Introduction

As a typical transition-metal oxide, titanium dioxide ($TiO_2$) has received extensive attention in recently several decades due to its widespread applications in the field of photocatalysis, dye-sensitized solar cells (DSCs), transparent conducting oxide (TCO) films, etc. [1–3]. In ambient conditions, it is well known that $TiO_2$ crystallizes in three representative polymorphs: rutile, anatase, and brookite. In light of its unique physicochemical characterizations with relatively high brightness, large refractive index (n = 2.75), chemical inertness, and large dielectric constant, rutile has been widely applied, such as in white pigment, opacifiers, and thin film capacitors [4,5].

A large quantity of high-pressure experimental and theoretical investigations has been employed to explore the phase stabilities and structural transitions for rutile by the synchrotron X-ray diffraction, Raman spectroscopy, and first-principles theoretical calculations. Previous results have already confirmed that there existed many high-pressure polymorphs for rutile, e.g., the columbite phase ($\alpha$-$PbO_2$, space group *Pbcn*) and the baddeleyite phase (MI, *P2$_1$/c*). However, till now, the high-pressure structural phase transition sequence and the pressure point of rutile to the baddeleyite phase transition has remained controversial. Some researchers think that rutile transformed directly to the baddeleyite phase without undergoing the intermediate phase of the columbite [6–12]. Furthermore, there exist

considerable disputes regarding the pressure point of rutile and the baddeleyite phase transition. Machon et al. [6] have investigated the Raman spectroscopy of rutile nanorods with a diameter of around 6–8 nm using a diamond anvil cell and revealed the phase transition of rutile and baddeleyite at a pressure of ~16.0 GPa. However, when the pressure was released, the new columbite phase appeared at ~0.2 GPa and remained stable in atmospheric conditions, whereas a similar study reported that synchrotron X-ray diffraction results on the rutile-to-baddeleyite phase transition for nanocrystalline rutile with an average grain size of 30 nm occurred at ~8.7 GPa by virtue of a diamond anvil cell [12]. Previous high-pressure Raman spectroscopy experiments in the diamond anvil cell have already confirmed that one available intermediate phase of columbite existed during the process of the phase transformation between rutile and baddeleyite at ~10.4 GPa with an initial grain diameter of 20–30 um, and further, the baddeleyite phase appeared at ~20.0 GPa [13].

As usual, the pressure-induced structural phase transition, metallization, and amorphization are accompanied by the variation of electrical transport characteristics for some engineering materials [14–18]. To the best of our knowledge, only one high-pressure electrical resistivity experiment on the synthetic rutile with various Ni-doped concentrations was reported under a limited pressure range by using a Bridgman opposed anvil setup [19]. They observed that the electrical resistivity of sample decreased drastically under the conditions of 4.0 GPa and 500 °C, and then became constant at the pressure range of 4.0–8.0 GPa, which indicated the occurrence of the semiconductor-to-metal phase transition in synthetic rutile. As for the nanocrystalline rutile, no relative high-pressure electrical transport properties have been reported so far. Therefore, a systematic study on the electrical transport characteristic for the nanocrystalline rutile is crucial under high pressure.

In the present work, we report two structural phase transitions and metallization for nanocrystalline rutile at pressures of up to ~25.0 GPa using the diamond anvil cell in conjunction with Raman spectroscopy, *AC* complex impedance spectroscopy, and high-resolution transmission electron microscopy. Furthermore, two correspondent structural phase transitions and metallization for nanocrystalline rutile under high pressure are discussed in detail.

## 2. Experimental Procedure

Natural rutile with a gem-class single crystal was gathered from Xinyi city, Jiangsu province, China. The single crystal was crushed and ground into the fine particles in an agate mortar. X-ray diffraction (XRD) analysis of the starting sample was collected by an X'Pert Pro X-ray powder diffractometer (Phillips Company, Amsterdam, The Netherlands, the Cu K$\alpha$ radiation with working voltage 45 kV and applied current 40 mA, respectively). Selected X-ray diffraction pattern was used to determine the lattice parameters of the starting sample by a Rietveld refinement as implemented in MDI Jade 6.5 software. Figure 1 shows the X-ray diffraction pattern of the starting sample; the observed XRD peaks are in good accordance with the tetragonal rutile in ambient conditions (space group: *P4$_2$/mnm*, JCPDS no. 88-1175). Some lattice parameters of rutile were calculated to be $a = b = 4.5933$ Å, $c = 2.9592$ Å, $\alpha = \beta = \gamma = 90°$, and $V = 62.43$ Å$^3$, which is close to the values in the International Centre for Diffraction Data (ICDD). The average particle size of the starting sample was calculated to be 72 nm by virtue of the Scherrer's equation, which is in good agreement with the result from the TEM observation (Figure S1).

High-pressure Raman spectroscopy measurements were performed using a DAC with an anvil culet of 300 μm. The ruby single crystal with its grain size of ~5 μm was applied to calibrate the pressure based on the shift of R1 photoluminescence line. To produce a hydrostatic environment, Helium was used as the pressure medium. Raman spectra were carried out using a Raman spectrometer (Invia, Renishaw, Wharton Anderch, UK) equipped with a confocal microscope (TCS SP8, Leica, Wetzlar, Germany) and a CCD camera (Olympus, Shinjuku, Tokyo, Japan). Spectra were taken in the backscattering geometry using an Argon ion laser (Spectra physics: 514.5 nm and power <1 mW) in the frequency shift range of 100–1000 cm$^{-1}$ with a spectral resolution of 1.0 cm$^{-1}$. Each spectrum was collected for 450 s. To avoid pressure oscillation, the equilibrium time of 15 min was kept at each

designated pressure point. The positions of Raman peaks were determined by fitting a Lorenz-type function using PeakFit software. The particle size and microstructure observations for the starting and recovered samples were investigated by the high-resolution transmission electron microscopy (HRTEM, Tecnai G2 F20 S-TWIN TMP, FEI, Hillsboro, OR, USA).

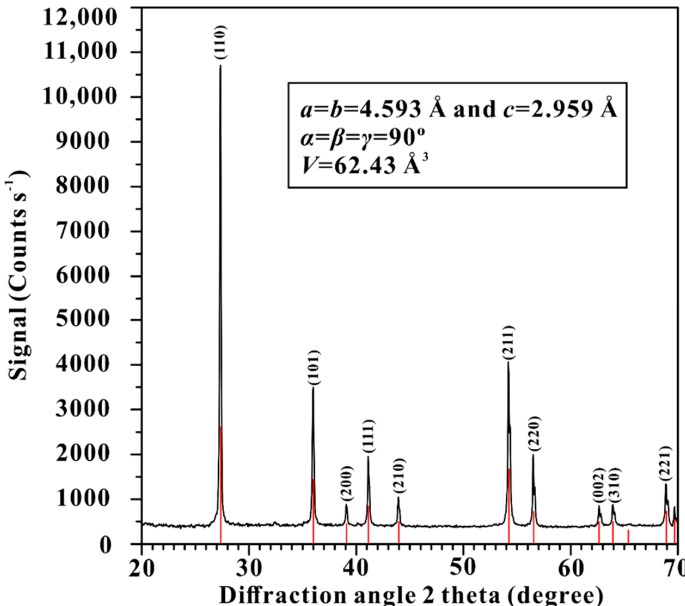

**Figure 1.** The X-ray diffraction (XRD) pattern of nanocrystalline rutile under ambient conditions.

High-pressure electrical conductivity experiments were conducted by a DAC with the anvil culet of 300 μm diameter. A T-301 stainless steel gasket was pre-indented into a thickness of ~40 μm, and a 180 μm center hole was drilled by a laser. Then, a mixture of boron nitride powder and epoxy resin was compressed into the hole, and another one 100 μm center hole was drilled as the insulating sample chamber. The *AC* complex impedance spectroscopy was measured using a Solartron-1260 impedance/gain phase analyzer in the frequency range of $10^{-1}$–$10^7$ Hz. The plate electrode was integrated into both diamond anvils. A low temperature was obtained by liquid nitrogen and an experimental temperature was measured by a *k*-type thermocouple with an estimated accuracy of 5 K. Detailed descriptions of the high-pressure experimental procedures and measurement methods can be found elsewhere [14–18].

## 3. Results and Discussion

High-pressure Raman spectroscopy was performed to investigate the structural property of nanocrystalline rutile at room temperature up to ~25.0 GPa. In Figure 2a, four typical Raman vibration modes for nanocrystalline rutile are observed in ambient conditions, which can be assigned as 143 cm$^{-1}$ ($B_{1g}$), 242 cm$^{-1}$ (multi-phonon), 441 cm$^{-1}$ ($E_g$), and 609 cm$^{-1}$ ($A_{1g}$). The peaks at 143 cm$^{-1}$ ($B_{1g}$) and 609 cm$^{-1}$ ($A_{1g}$) are related to the O-Ti-O bond bending and Ti-O bond stretching modes, while the 441 cm$^{-1}$ ($E_g$) peak is due to the oxygen atom liberation along the c-axis orientation [20]. An anomalously strong and broad peak at 242 cm$^{-1}$ is a multi-phonon peak caused by the second-order Raman scattering experiment in rutile structure. All of these observed Raman characteristic peaks are in good agreement with previous studies in ambient conditions [21,22]. At the pressure range of 0–12.3 GPa, all of the Raman peaks for rutile phase shifted toward higher frequencies with increasing pressure, except for the $B_{1g}$ soft mode. The red shift of the $B_{1g}$ soft mode, which is characterized by the negative pressure-dependent Raman peak, can provide a clue to the instability of rutile structure under high pressure. Our observed phenomenon of red shift in rutile phase also existed in some similar rutile-structured compounds, such as $SnO_2$ and $GeO_2$ [23]. At ~12.3 GPa, six acquired new peaks at

around 229, 278, 323, 445, 674, and 721 cm$^{-1}$ were identified as the baddeleyite phase [24–26], which demonstrated the occurrence of phase transition from rutile to baddeleyite phases. When the pressure was continuously enhanced up to 13.8 GPa, the Raman peak at 494 cm$^{-1}$ was split into two new separate peaks at 511 and 529 cm$^{-1}$, respectively. The splitting phenomenon in the baddeleyite phase was possibly related to the nanometer size effects [24]. Furthermore, the Raman peaks of baddeleyite phase shifted toward higher frequencies, which indicated the structure of baddeleyite phase remained stable up to the highest pressure of ~25.0 GPa.

The evolution of the Raman shift for nanocrystalline rutile under pressure (pressure coefficient, d$v$/d$P$) is plotted in Figure 2b. Two discrete pressure ranges can be identified by the variation of the slope of pressure coefficient: the pressure ranges from ambient to 12.3 GPa, and from 12.3 to 25.0 GPa, respectively. A discontinuous change in the pressure coefficient at ~12.3 GPa indicates the structural phase transition from rutile to baddeleyite phases. The fundamental structural units in rutile and columbite phases are of the TiO$_6$ octahedrons with totally different link modes. As for the high-pressure baddeleyite phase, each tetravalent titanium cation (Ti$^{4+}$) is coordinated with seven divalent oxygen anions (O$^{2-}$) and forms the distorted fluorite structure [27]. A discontinuous change in the pressure coefficient at ~12.3 GPa arises from the distortion and breakdown of TiO$_6$ octahedron during the process of phase transition. Thus, the occurrence of phase transition from rutile to baddeleyite phases is possibly related to the variation of coordination number in the tetravalent titanium cation (Ti$^{4+}$) [28].

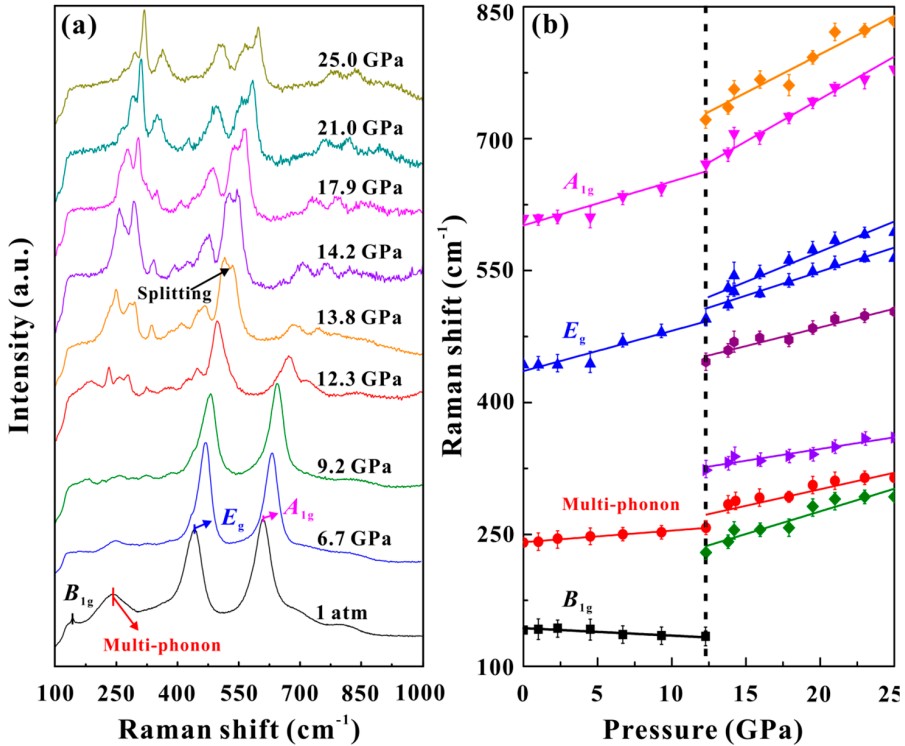

**Figure 2.** (**a**) Raman spectra of nanocrystalline rutile at representative pressures during compression process. (**b**) The evolution of the Raman shift with increasing pressure (d$v$/d$P$) at atmospheric temperature.

Upon decompression, the Raman peaks of baddeleyite phase continuously shifted toward lower frequencies in the pressure range of 25.0–7.2 GPa, as presented in Figure 3a. When the pressure was decreased to ~7.2 GPa, new Raman peaks appeared at the positions of 134, 230, 277, 320, 378, 442, 512, 645, and 706 cm$^{-1}$. All of these representative Raman peaks are the characteristic of the columbite phase [10,26,29,30], which suggests the occurrence of phase transition from baddeleyite to columbite phases at ~7.2 GPa. As the pressure was continuously reduced, all of these Raman intensities for the columbite phase became obviously stronger. Therefore, the phase transformations from rutile to baddeleyite to columbite phases were irreversible. The corresponding pressure-dependent Raman

shift of nanocrystalline rutile during decompression is detailedly illustrated in Figure 3b. The available inflexion point of the pressure coefficient at ~7.2 GPa displays the structural phase transition from baddeleyite to columbite phases, which is possibly related to the variation of coordination number in the tetravalent titanium cation ($Ti^{4+}$) [28].

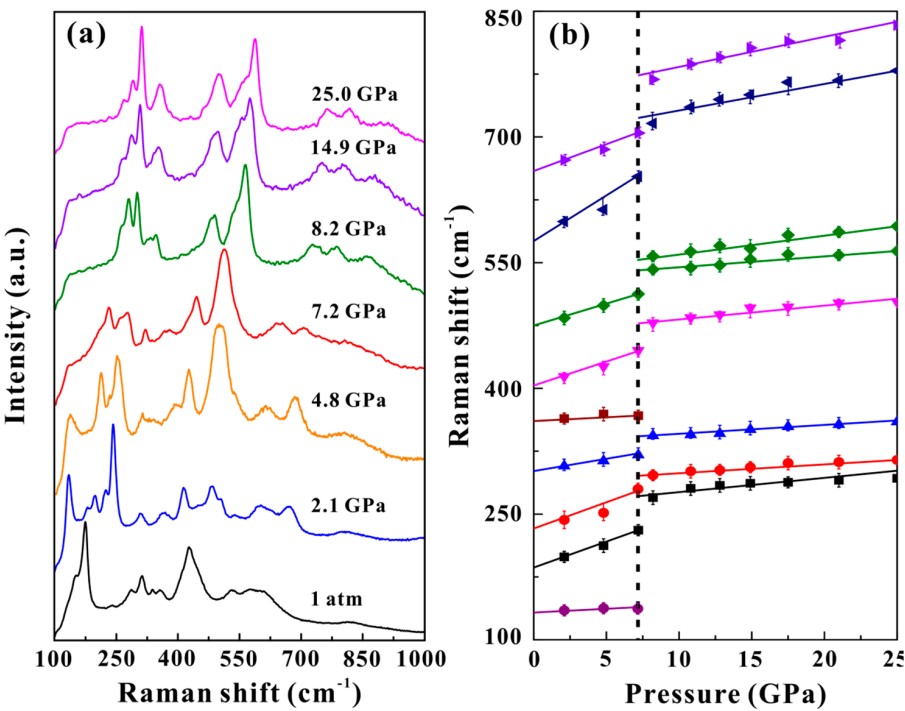

**Figure 3.** (**a**) Raman spectra of nanocrystalline rutile at selected pressures in the process of decompression. (**b**) The pressure dependence of the Raman shift ($dv/dP$) at room temperature.

The representative Nyquist diagrams of the impedance spectroscopy for nanocrystalline rutile at atmospheric temperature during compression are displayed in Figure 4a–c. At the pressure range of 1.6–12.3 GPa, the impedance spectra exhibit a semicircle within the high-frequency range and a low-frequency oblique line. Each impedance semicircular arc was fitted by the equivalent circuit consisting of a parallel resistor (R) and constant-phase element (CPE). Further increasing the pressure, the grain boundary effect of the sample became weaker gradually. When the pressure was higher than 14.5 GPa, the impedance arc only appeared in the fourth quadrant, and it could be fitted only by the simple resistor (R). The representative Nyquist diagrams of the impedance spectroscopy for baddeleyite and columbite phases upon compression are presented in Figure S2. Only one impedance semicircular arc of grain interior or one pure resistance was obtained among the phases of baddeleyite and columbite. The electrical conductivities of the samples can be calculated as follows:

$$\sigma = L/SR \tag{1}$$

where $L$ is the distance between the two electrodes (cm), $S$ is the cross-sectional area of the electrode ($cm^2$), $R$ is the resistance of sample ($\Omega$), and $\sigma$ is the electrical conductivity of sample (S/cm). Figure 4d shows the pressure-dependent electrical conductivity of the grain interior and boundary for the nanocrystalline rutile in the process of compression and decompression at atmospheric temperature. During compression, the electrical conductivity of grain interior increases with increasing pressure, and three linear regions were obtained on the base of various slopes. At the pressure ranges of 1.6–12.3 GPa and 14.5–25.0 GPa, the grain interior electrical conductivity enhances slowly with increasing pressure at the rates of 0.032 and 0.041 S $cm^{-1} \cdot GPa^{-1}$, respectively. However, the grain interior electrical conductivity increases drastically by about four orders of magnitude at 12.3–14.5 GPa. The grain

boundary electrical conductivity shows the opposite trend at the pressure range of 1.6–12.3 GPa, and then disappears above 12.3 GPa. The available discontinuities of electrical conductivity for both the grain interior and boundary at ~12.3 GPa are observed, which hint the occurrence of phase transition from rutile to baddeleyite phases. Above 14.5 GPa, the sample electrical conductivity within the range of 6–11 S cm$^{-1}$ may be indicative of metallization. The electrical conductivity magnitude remains constant at ~12 S cm$^{-1}$ at 25.0–7.2 GPa and then decreases within the range of 5–11 S cm$^{-1}$ below ~7.2 GPa. The available inflexion point at ~7.2 GPa is consistent with our above-mentioned Raman scattering results, which can be ascribed to the occurrence of transformation from baddeleyite to columbite phases. In a similar study, Olsen et al. [31] observed the structural phase transition from rutile to baddeleyite at a higher pressure range of 20–30 GPa with an average grain size of 10 nm. This was possibly related to the different grain size, which may have resulted in a discrepancy of the pressure point of phase transition and the width of the phase coexistence regime reported by Olsen et al. and us.

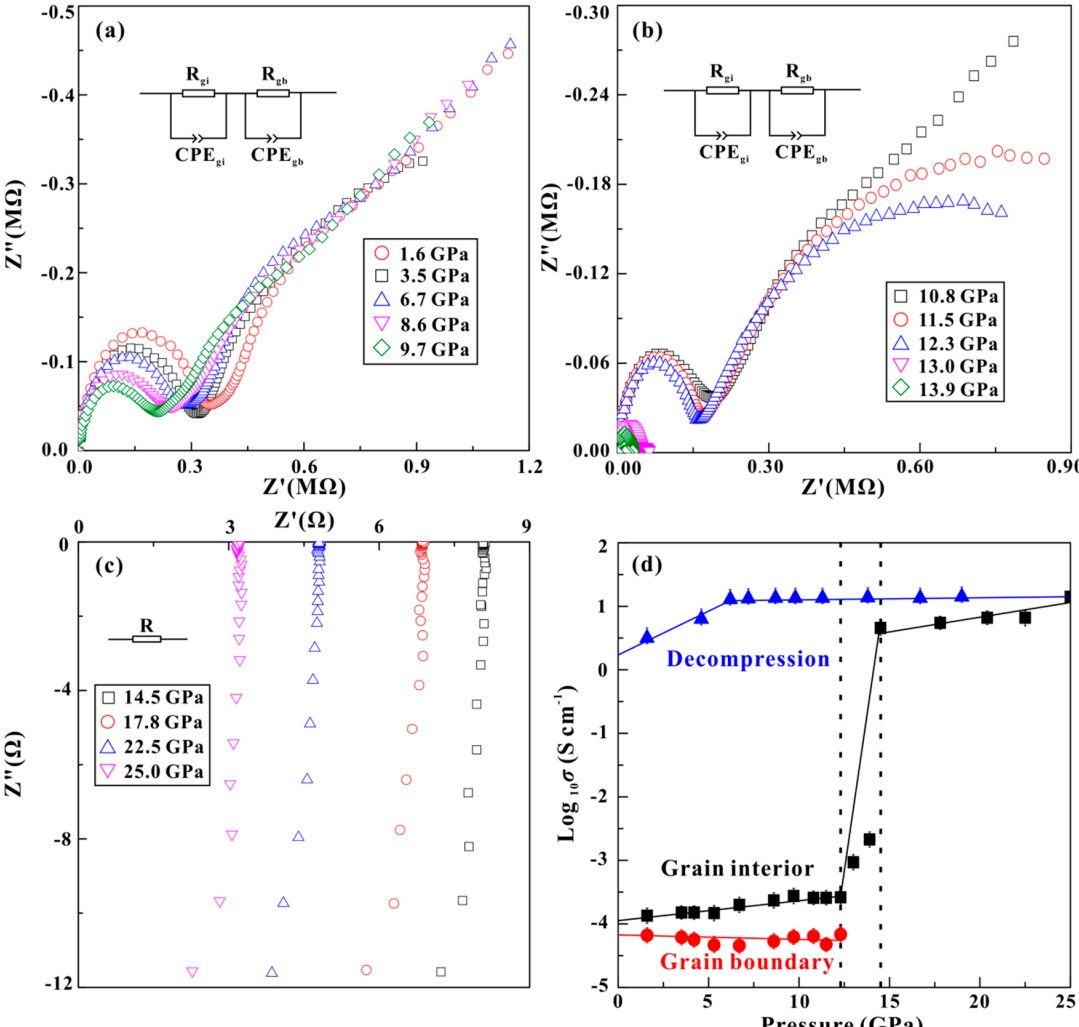

**Figure 4.** (**a**–**c**) Nyquist diagrams of the impedance spectra for nanocrystalline rutile at different pressures. $R_{gi}$ and $R_{gb}$ represent the resistance of the grain interior and boundary, respectively. $CPE_{gi}$ and $CPE_{gb}$ are the constant phase element of the grain interior and boundary, respectively. (**d**) The grain interior and boundary electrical conductivity of nanocrystalline rutile during compression and decompression process at atmospheric temperature.

To check the high-pressure metallization of nanocrystalline rutile, we performed temperature-dependent electrical conductivity measurements up to 25.0 GPa at 120–240 K. As usual, the electrical

conductivity of sample increased with increasing temperature for semiconductor, whereas the metal exhibited a negative relation between the temperature and electrical conductivity [15–18]. The temperature-dependent electrical conductivity measurements of nanocrystalline rutile at selected pressures are plotted in Figure 5. Below 13.2 GPa, the electrical conductivity of sample increases with increasing temperature, displaying a typical characterization of semiconductor. A negative relation between electrical conductivity and temperature above 14.8 GPa indicates the occurrence of metallization. At 0.3 GPa, the recovered columbite phase also shows a typical metallic behavior. As usual, there are two dominant causes for the occurrence of metallization phenomenon in semiconducting materials: the closure of bandgap and the drastic increase of defect concentration under high pressure. In order to effectively distinguish the metallization mechanism in our present rutile sample, first-principles theoretical calculations were implemented to predict the electronic and structural evolutions of rutile and baddeleyite phases under high pressure in the Supplementary Information (Figures S3 and S4). This made it clear that the bandgap energy of rutile phase fells within the range of 1.99 eV to 1.96 eV when the pressure increased from 0 GPa to 12.0 GPa. As for the baddeleyite phase, the bandgap energy fells within the range of 2.21 eV to 2.17 eV when the pressure increased from 14.0 GPa to 25.0 GPa. Therefore, it is impossible that the occurrence of pressure-induced metallization for rutile is related to the closure of bandgap. An absolutely new experiment was performed to observe the variation of color for nanocrystalline rutile under high pressure using a diamond anvil cell, as shown in Figure S5. We found that there was no observable color change at the pressure range of 0–10.0 GPa. However, when the pressure increased to 15.0 GPa, one obvious variation of color in sample from almost white to black (dark) transition was observed. As a matter of fact, the colors from the shallow to deep variation in rutile stand for the enhancement of oxygen vacancies concentration in $TiO_2$ particles [32]. Therefore, the obvious color variation of nanocrystalline rutile is strongly related to the enhancement of the defect concentration under pressure, which results in the occurrence of metallization. Therefore, the metallization of nanocrystalline rutile is attributed to the enhancement of defect concentration rather than the closure of bandgap.

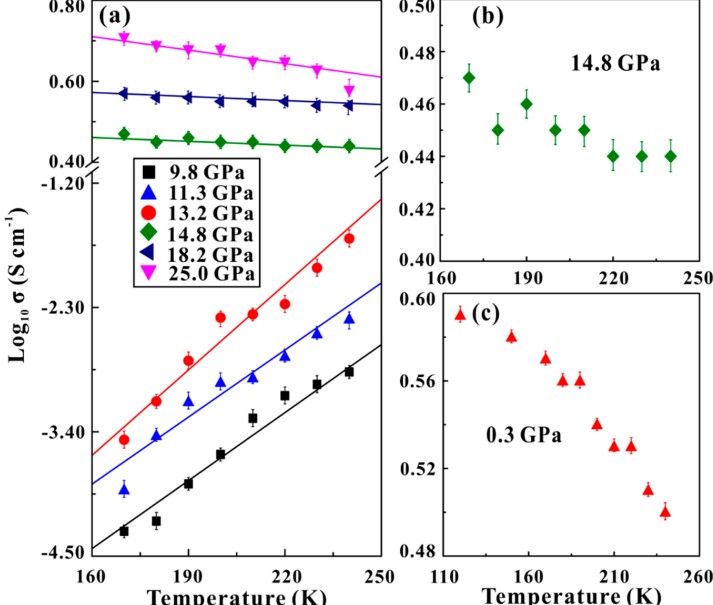

**Figure 5.** (**a**) The pressure dependence of electrical conductivity for nanocrystalline rutile as a function of temperature, the highest pressure achieved in the experiment is 25.0 GPa. The equilibrium time of 20 min was kept at each designated pressure point. (**b**) The metallic state of baddeleyite phase at ~14.8 GPa during compression process. (**c**) The metallic property of columbite phase after quenched down to 0.3 GPa. A relatively longer equilibrium time of 120 min was applied at almost atmospheric pressure in order to decrease the experimental uncertainty of electrical conductivity for sample.

In order to further investigate the reversibility of the structural phase transition for nanocrystalline rutile, HRTEM observation was performed for both of the starting and recovered samples. In initial HRTEM image in Figure 6a, the interplanar distance value is ~0.32 nm, which corresponds to the (110) plane of rutile phase. At the same time, the initially selected area electron diffraction (SAED) pattern (Figure 6c) consists of a series of rings with bright discrete diffraction spots, which can be identified as rutile phase. In Figure 6b of the recovered HRTEM image, the interplanar distance values are ~0.27 and ~0.35 nm, assigned to the (020) and (110) planes of the columbite phase, respectively. Meantime, the corresponding SAED of the recovered sample exhibits a few clear spots, which were confirmed as a columbite phase [33]. Thus, the nanocrystalline rutile eventually transformed and maintained the columbite phase under ambient conditions. In conclusion, all of these obtained results on nanocrystalline rutile from the Raman spectroscopy experiments and HRTEM observations revealed the irreversibility of the structural transformation under pressure.

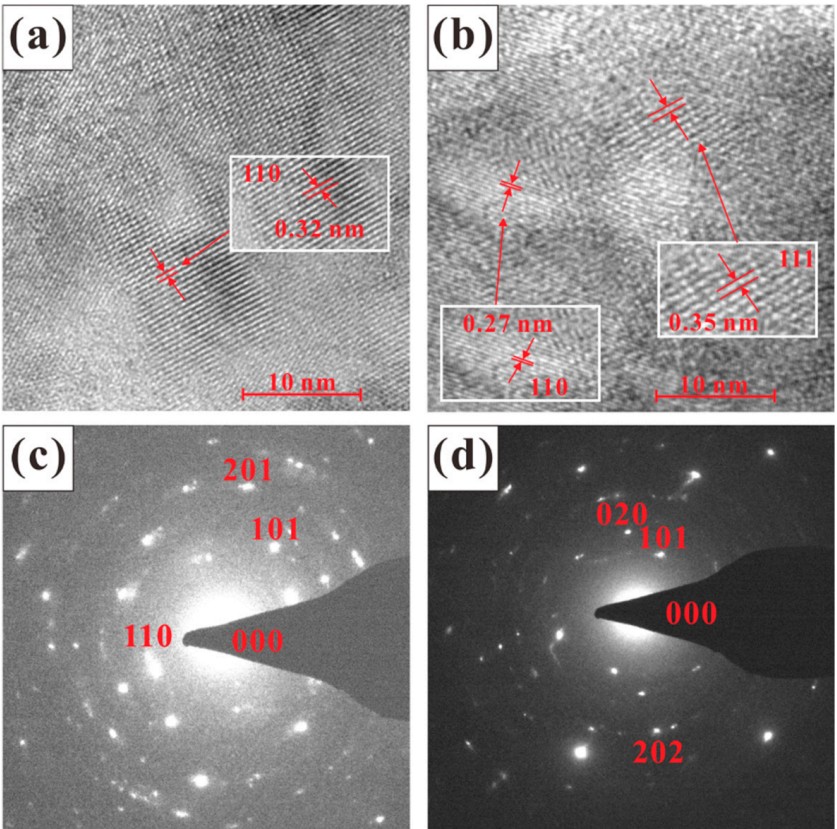

**Figure 6.** (**a,b**): high-resolution transmission electron microscopy (HRTEM) images of the starting and recovered samples, respectively. (**c,d**): the corresponding selected area electron diffraction (SAED) patterns.

## 4. Conclusions

We have reported two structural phase transitions and metallization for nanocrystalline rutile using the diamond anvil cell at around 7.2, 12.3, and 14.5 GPa, respectively. During compression, the structural phase transition from rutile to baddeleyite phases at ~12.3 GPa was disclosed by the appearance of new characteristic peaks in Raman spectroscopy, the inflexion point of the pressure coefficient, and pressure-dependent electrical conductivity. As the pressure was continuously increased up to ~14.5 GPa, the high electrical conductivity value provided a crucial clue regarding metallization, which was confirmed by the temperature-dependent electrical conductivity measurements. Upon decompression, the pressure-dependent Raman peaks and electrical conductivity for the columbite phase indicated the occurrence of structural phase transformation from baddeleyite to columbite

phases at ~7.2 GPa. The HRTEM observations on the starting and recovered samples demonstrated that the phase transformations from rutile to baddeleyite to columbite phases were irreversible under high pressure.

**Supplementary Materials:** The following are available online at http://www.mdpi.com/2075-163X/9/7/441/s1, Figure S1: (a) and (b) are the TEM images of the starting sample. (c) and (d) the corresponding histograms of the particle size distribution. It is one of the potentially effective and good methods that the TEM observation can be used to determine the particle size distribution state in our starting sample. As shown in Figure S1 (a) and (b), the starting rutile particles with almost homogenous distribution state. We estimated roughly that there existed at least 20 and 8 particles in Figure S1 (a) and (b), respectively. Figure S1 (c) and (d) represent the corresponding histograms of the particle size distribution for the starting sample, most of the particle size are within the range of 70–80 nm. The average particle size of the starting sample was estimated to be 78 nm, which is in good consistent with the result from XRD; Figure S2: (a) The Nyquist diagram of the impedance spectra for baddeleyite phase at the pressure range of 19.0–8.7 GPa during decompression. (b) The Nyquist diagram of the impedance spectra for columbite phase at the pressure range of 7.2–1.6 GPa during decompression, the equivalent circuit of R stands for the resistance. Figure S3: (a) and (b) Calculated band structure for rutile phase at the pressures of 0 GPa and 10.0 GPa. The bandgap energy for rutile phase are 1.99 eV and 1.96 eV at the pressures of 0 GPa and 10.0 GPa, respectively. (d) and (e) The corresponding total density and projected density at the pressures of 0 GPa and 10.0 GPa for rutile phase. (c) Calculated band structure for baddeleyite phase at 25.0 GPa. The bandgap energy for baddeleyite phase is 2.17 eV at 25.0 GPa. (f) The corresponding total density and projected density at 25.0 GPa for baddeleyite phase. Figure S4: The calculated bandgap energy of rutile phase at the pressure range of 0–12.0 GPa and the baddeleyite phase within the pressure range of 14.0–25.0 GPa. Figure S5: (a) The optical microscope image of the starting material for nanocrystalline rutile. (b) and (c) The optical microscope images of the nanocrystalline rutile at the pressure points of 10.0 GPa and 15.0 GPa using the diamond anvil cell, respectively.

**Author Contributions:** L.D. designed the project. M.H. and L.D. wrote the initial draft of the work and the final paper). M.H., L.D., H.L., H.H., K.L., L.Y., and C.P. interpreted the results. L.D. corrected and recognized the final paper. M.H. and K.L. performed and interpreted the high-P experiments and the HRTEM images. All authors discussed the results and commented on the manuscript.

**Funding:** This research was financially supported by the strategic priority Research Program (B) of the Chinese Academy of Sciences (18010401), Key Research Program of Frontier Sciences of CAS (QYZDB-SSW-DQC009), Hundred Talents Program of CAS, NSF of China (41774099 and 41772042), Youth Innovation Promotion Association of CAS (2019390), Special Fund of the West Light Foundation of CAS, and Postdoctoral Science Foundation of China (2018M643532).

**Acknowledgments:** We thank the editor for kindly handling our paper, as well as two anonymous reviewers for their constructive and enlightened advice in the revising process.

**Conflicts of Interest:** The authors declare no conflict of interest.

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
