# Peer review of "Structural Phase Transition and Metallization of Nanocrystalline Rutile Investigated by High-Pressure Raman Spectroscopy and Electrical Conductivity"

_minerals, doi:10.3390/min9070441_

Round 1

Reviewer 1 Report

Hong et al. submitted their manuscript dedicated to the high-pressure behavior of nanocrystalline rutile TiO2 after revisions. I appreciate the new information they are providing with XRD and TEM analysis, mainly the particle size and the size distribution.

I have few minor comments and one major.

First, concerning the Raman spectroscopy measurements:

-          L.114 I do not understand the term “liberation”

-          The band centered at 242 cm-1 is often interpreted as a multiphonon process. This point has been recently discussed by Machon et al (J. Phys Chem C123, 1948 (2018) ref[6] in the manuscript). Can the authors comment on this point? For instance, is there an effect of the preparation method on the defect density (grinding a single crystal).

The Raman data are in agreement with the literature and do not bring any new insights.

The results obtained on the electrical conductivity are much more originals. However, they rise a question that needs to be answered before publication:  to the best of my knowledge, there is no report of intrinsic metallization of TiO2 under pressure : no report of any change of color, the Raman spectra remain as expected for a semiconductor (no Fano profile, no drastic decrease of the scattering intensity) and ab initio simulations indicate that baddeleyite is a semiconductor with a gap between 2 and 4 eV. The authors refer to another very recent paper on anatase TiO2 showing resistive switching effect (ref 30). In my opinion, the mechanism at work must be discussed more extensively with an emphasis on the effect of defects. Also, a discussion on what happened with the grain boundaries would be welcome.  

In other words, such surprising results must be discussed and commented in a more explicit way than just a reference to a recent paper.

Reviewer 2 Report

The manuscript describes a thorough study of the pressure-induced phase transformations in rutile TiO2 upon compression and decompression through in situ Raman and electrical impedance measurements. In general the study is well-described, cohesive and presents a useful and compelling set of results that builds upon current work in the literature. I recommend publication of this manuscript after minor revisions to contextualize the measured results with the literature, and perhaps general edits to correct minor English language errors. Below are specific comments:

1.       In the introduction, the authors present the controversy in determining the precise pressures at which the rutile to baddeleylite, and baddeleylite to columbite transitions occur. It would be helpful for the reader if the authors could place their measured values in context with earlier results. In particular, can the authors comment on what differences in measurement apparatus or sample prep would cause discrepancies in the measured value of the phase transition pressures by in situ high-pressure Raman experiments from earlier works? It seems as though nanomaterial grain size changes the transition pressure from the cited works in the introduction, so perhaps this may explain some of the differences in measured transformation pressures. Similarly, how does the width of the phase coexistence regime during compression (i.e. between 12.3 and 14.5 GPa) compare to previous studies? Does this also indicate differences in sample preparation from previous Raman or diffraction measurements? There does not appear to be any intermediate columbite phase forming during this transition based on the narrative of the paper, and it would be helpful to place this result in context with earlier works that find different results.

2.       There appears to be hysteresis in the Raman shift vs. Pressure plots shown in Figs. 2b and 3b. The dv/dP slope seems to be much steeper upon compression than decompression. Is there a simple explanation for this result? Earlier studies (e.g. Gerward and Olsen) didn’t seem to observe much hysteresis in the high-pressure regime, so perhaps this result also points to the effects of sample preparation.

3.        The impedance data is compelling and agrees well with the Raman measurements. Could the authors also include representative Nyquist diagrams upon decompression, perhaps in the SI, to support the presented conductivity values in Figure 4d?

Author Response

This manuscript is a resubmission of an earlier submission. The following is a list of the peer review reports and author responses from that submission.

Round 1

Reviewer 1 Report

The article by Hong et al reports phase transition in nanocrystalline rutile by Ramna spectroscopy and electrical condutivity at high pressure. The structural phase tranformation seems to be accompanied by a metallization of the sample.

This work has currently very little interest. The basic hypothesis is that the sample is nanocrystalline. There is no proof of this claiming. The X-Ray diffraction (Fig. 1) may be used to determined the particle size using at least the Scherrer equation (I would prefer a Williamson-Hall plot as the powder is ground). TEM may also be used to obtain the size distribution of the particles. 

Usually, in TiO2 nanoparticles , the observed phase is the anatase one.

Without these basic information, the paper has nearly no value.

I cannot recommend this paper for publication. 

Reviewer 2 Report

The authors investigated the structural phase transitions of a TiO2 rutile powder under pressure using in situ Raman and AC complex impedance spectroscopies. They observed a semiconductor-metal transition above 14 GPa corresponding to the transition rutile-baddeleyite. During decompression, the baddeleyite phase transforms to a metallic columbite phase. This sequence of phase transitions has already been observed using X-ray diffraction or Raman spectroscopy and reported by different research groups. The novelty of the present work is based on the observation of the metallization of TiO2 above 14GPa. This result is really surprising and in opposition with the resistivity measurement performed under pressure on the baddeleyite and columbite phasis reported in the paper ( X. Lü, W. Yang, Z. Quan, T. Lin, L. Bai, L. Wang, F. Huang, and Y. Zhao, Journal of the American Chemical Society 136, 419 (2014)). In this former work on Nb doped TiO2 (not cited by the present authors) no transition semiconductor-metal is observed. The pressure cycle up to 40 GPa only induced an increase of 40% of the conductivity, but the recovered sample was still a semiconductor.

The results and discussion of the measurements with the impedance spectroscopy have to be improved:

The conductivity given in figure 4(d) for TiO2 at ambient temperature and low pressure at the beginning of the pressure cycle is about 3.10-5 S.cm-1. This value can be compared to the conductivity measured on a TiO2 rutile paste, 2.10-6 S.cm-1 (Mohd et al., Journal of Physics: conference series, 495 (2014) 012027 (doi: 10.1088/1742-6596/495/1/012027)). At 9,8 GPa the conductivity of the rutile (Fig.4(d)) is about 10-4 S.cm-1, but on the figure 5 (a), if we extrapolate the curve (rutile 9,8 GPa) to the ambient temperature we find a conductivity close to 2,5.10-2 S.cm-1. This big difference suggests an error in the reported values. This should be clarified.

In the figure 5(c) the error bars are surprisingly small compared to the error bars given in the upper part of Fig. 5 (a).

The Fig 5 (b) is not useful. The same curve is already on Fig 5 (a). It may be preferable to plot the upper part 5(a) at this place instead.

Why the authors did not separate the conductivities extracted for grain interior and grain boundaries as they did in their previous paper (C. Pu et al., Spectroscopy letters, 51 (2018) 10, 531-39)?

The grain size of the powder is not given. The XRD peaks appear very sharp for a nanopowder.

The present paper needs major revision of the interpretation of the measurements obtained with the AC complex impedance spectroscopy, before a reconsideration.